# Test-Time Adaptation of Vision-Language Models with Low-Rank Pseudo-Consistency

**Shuvendu Roy**                                                    *shuvendu.roy@queensu.ca*
*Queen's University, Canada*

**Ali Etemad**                                                        *ali.etemad@queensu.ca*
*Queen's University, Canada*

**Reviewed on OpenReview:** *https://openreview.net/forum?id=GDw4pvX9aG*

## Abstract

While test-time adaptation (TTA) methods enable vision-language models (VLMs) to adapt under distribution shifts, they typically rely on simple feature transformations following frozen encoders while learning from potentially noisy pseudo-labels. This approach may limit adaptation under significant domain shifts. In this paper, we propose PseudoAdapter, a novel TTA framework for VLMs that introduces low-rank adapters into early layers of the encoder to enable domain-specific feature adaptation while maintaining generalization. To ensure effective learning from noisy and low-confidence predictions, PseudoAdapter combines confidence-calibrated pseudo-labelling with unsupervised consistency learning across augmented views. We further extend our approach with PseudoAdapter+, which integrates selective teacher supervision to improve adaptation with minimal overhead. Extensive evaluations on four out-of-distribution and ten cross-domain benchmarks demonstrate that our method outperforms prior state-of-the-art TTA approaches by an average of 6.84% and 3.25%, respectively. Ablation studies confirm the effectiveness of each proposed component.

## 1 Introduction

Vision-language models (VLMs) (Yang et al., 2022; Jia et al., 2021; Li et al., 2021; 2022; 2023; Gao et al., 2024) have demonstrated remarkable generalization and state-of-the-art (SOTA) performance across diverse downstream tasks, such as image recognition (Wang et al., 2023; Wasim et al., 2023) and generation (Kumari et al., 2023; Rombach et al., 2022; Guo et al., 2024). By being trained on large-scale image-text datasets, VLMs learn to align representations across modalities, which enables zero-shot classification without task-specific fine-tuning. However, in real-world settings, the test data distribution often differs significantly from the pre-training data, which often leads to suboptimal performance. To address this, test-time adaptation (TTA) methods (Wang et al., 2020; Iwasawa & Matsuo, 2021; Shu et al., 2022; Karmanov et al., 2024) have been proposed, which align VLMs to unseen distributions during inference. These methods include training-free approaches (Iwasawa & Matsuo, 2021; Zhang et al., 2022a; Karmanov et al., 2024) and training-required techniques (Wang et al., 2020; Shu et al., 2022; Samadh et al., 2023; Zhang et al., 2022b), with recent advancements primarily focusing on the latter.

We identify two key limitations in current TTA approaches: (1) Most methods, such as (Zhang et al., 2022b; Karmanov et al., 2024), rely on learning simple feature transformations (e.g. linear probing or prototype learning) on top of a frozen pre-trained encoder (Figure 1 (left)) to address distribution shifts. However, the representations captured through this approach may not remain effective across distributions that differ substantially from that of the training data. (2) SOTA TTA methods (Zhang et al., 2022b; Karmanov et al., 2024) primarily utilize pseudo-labels generated by the pre-trained encoder for learning the test distributions. However, our preliminary empirical findings (see Figure 2) reveal that even pseudo-labels with high confidence are frequently noisy or incorrect, thus hindering the effectiveness of adaptation. Additionally, this approach

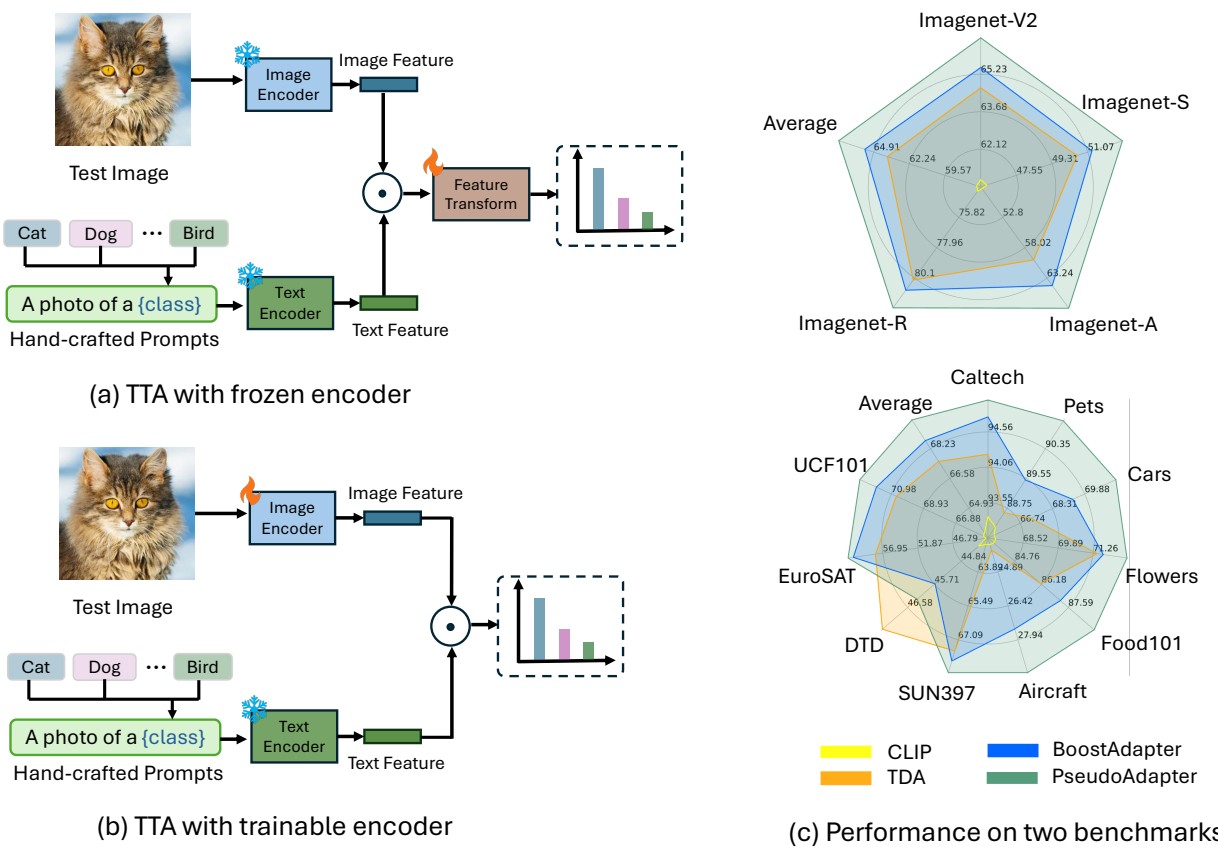

Figure 1: (a) Most existing SOTA TTA methods learn feature transformations following a frozen pre-trained encoder, which may not remain effective across strong distribution shifts. (b) Our proposed PseudoAdapter carefully tunes the pre-trained encoder for TTA. (c) PseudoAdapter shows considerable improvement over existing TTA methods.

only leverages high-confidence samples for learning, missing the opportunity to utilize the remaining samples, as no samples are stored or revisited in the TTA setup.

To address the above-mentioned problems, we propose a novel test-time adaptation method for VLMs called pseudo-consistency learning with low-rank adapters (**PseudoAdapter**). Instead of solely learning a feature transformation on a frozen encoder, our method enables the learning of domain-specific features by fine-tuning the encoder using low-rank adapters (LoRA). However, such fine-tuning could lead to overfitting, especially in low-shot settings like test-time adaptation. To mitigate this, our method strategically inserts adapters only into the early layers of the encoder, allowing the model to adapt to test-time features without losing its generalization capability. Furthermore, instead of solely learning from pseudo-label predictions and relying only on high-confidence samples, we propose a pseudo-consistency learning approach that (a) utilizes the entire test set for learning the target domain, and (b) mitigates the impact of erroneous pseudo-labels. Specifically, we use high-confidence samples in a supervised setting by treating their pseudo-labels for supervision. However, at the early stages of adaptation, pseudo-label predictions are often noisy due to miscalibration, leading to incorrect predictions with high confidence. To address this, we introduce a confidence calibration module that maintains a very high threshold for pseudo-label acceptance at the start of adaptation and gradually lowers the threshold as the model adapts to the target domain. Meanwhile, to also learn from samples with low-confidence predictions, we enforce consistency across augmented views of the same input, enabling learning in an unsupervised manner. This hybrid approach ensures that all test-time samples contribute to adaptation while minimizing the risk of incorrect supervision. To further enhance the adaptation of the pre-trained encoder, we introduce **PseudoAdapter+**, a teacher-guided adaptation

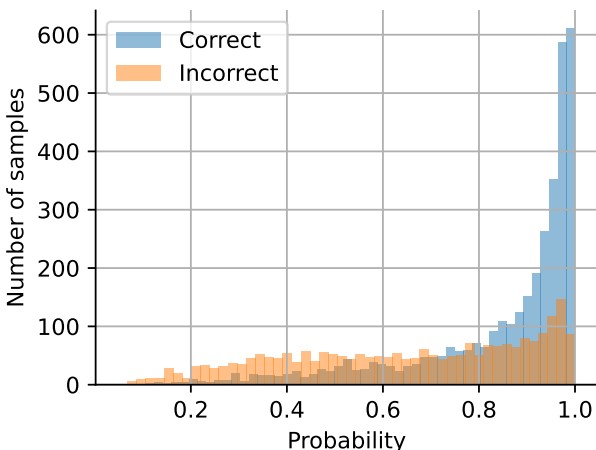

Figure 2: Histogram of pseudo-label prediction probabilities for CLIP-B/16 on ImageNet-A (1000 classes): correct predictions (blue) vs. incorrect predictions (orange), computed over the full test set. Even among predictions with softmax probability $\geq 0.9$, a substantial fraction are incorrect, illustrating the miscalibration of the pre-trained VLM under distribution shift and motivating our confidence calibration module.

strategy that selectively gathers teacher supervision to improve test-time adaptation on the target domain. The selective teacher supervision ensures that the overall computational budget of the adaptation pipeline does not increase significantly (under 5% overhead). We treat PseudoAdapter+ as a variant of our proposed method and present final results for both the base method (without teacher guidance) and PseudoAdapter+.

To evaluate PseudoAdapter we follow the established evaluation protocol of existing methods such as (Zhang et al., 2022b; Karmanov et al., 2024), and present the results on four OOD benchmarks: ImageNet-V2, ImageNet-Sketch, ImageNet-A, ImageNet-R, and ten cross-domain benchmarks: Caltech, Oxford Pets, Stanford Cars, Oxford Flowers, Food101, FGVCAircraft, SUN397, EuroSAT, and UCF101. On average, our proposed method shows 6.84% improvement over the existing SOTA in the OOD benchmarks, and a 3.25% average improvement on the cross-domain benchmarks (Figure 1 (right)). Extensive ablation and sensitivity studies show the effectiveness of each component of our method.

***How is PseudoAdapter different from other TTA methods?*** Low-rank adaptation has been previously explored for adapting vision-language models (Imam et al., 2025), but no prior work has investigated its use for test-time adaptation of VLMs to capture target domain features specifically in the early layers of a pre-trained encoder, and hasn't achieved balanced adaptability and retention of the generalization of the pre-trained model. Similarly, no existing work has combined pseudo-label supervision with unsupervised consistency learning to ensure effective utilization of all samples from the target domain. Additionally, our confidence calibration module is a novel contribution that mitigates noisy pseudo-label generation caused by miscalibration in the pre-trained VLM. Finally, while dual-encoder designs involving a teacher are commonly used for knowledge distillation, our teacher supervision strategy differs by selectively utilizing teacher supervision only for very low-confidence samples, thereby maintaining efficiency without significantly increasing computational cost. Overall, our method improves performance over existing TTA methods by an average of 6.84%.

Overall, we make the following contributions in this paper:

- To address the limitations of existing TTA methods, we propose PseudoAdapter, a novel approach that introduces low-rank adapters into early encoder layers for learning effective representations, and then combines pseudo-label supervision (with confidence calibration) and unsupervised consistency learning to benefit from the information within all the test samples in the target domain.

- We introduce a variant of our method, PseudoAdapter+, which incorporates a selective teacher supervision strategy to further improve TTA.

- Extensive evaluations demonstrate that our proposed solution outperforms all existing TTA methods across multiple benchmarks. To enable rapid reproducibility and contribute to the field of TTA, we will make our code public upon acceptance.

## 2 Related Work

Vision-language models (VLMs) have demonstrated remarkable generalization capabilities through contrastive pre-training on large-scale image-text datasets (Jia et al., 2021; Radford et al., 2021; Li et al., 2022; 2023). A notable example is CLIP (Radford et al., 2021), which is trained on 400 million image-caption pairs to align visual and textual representations. A substantial body of research has explored adapting CLIP to downstream tasks with limited labelled data, showing strong generalization performance (Zhou et al., 2022b;a; Zhang et al., 2022a; Li et al., 2024; Zhu et al., 2023; Lu et al., 2023). For instance, CoOp (Zhou et al., 2022b) introduces trainable prompts (Lester et al., 2021; Zha et al., 2023; Yang et al., 2024; Liu et al., 2021) to fine-tune CLIP, while CoCoOp (Zhou et al., 2022a) improves generalization by conditioning prompts on image embeddings. Similarly, MaPLe (Khattak et al., 2023) proposes a dual-branch prompting strategy for both vision and language modalities, refining cross-modal alignment. While these methods achieve strong performance, they depend on labelled target-domain data for adaptation, making them unsuitable for many practical applications where test distribution often deviates from the training data distribution.

More recently, test-time adaptation methods have been proposed that aim to adapt pre-trained VLM to the target distribution during inference. TTA methods can be broadly categorized into two categories: training-required and training-free methods. Training-free TTA approaches employ non-parametric techniques, such as cache models or prototypes, to predict test sample labels without model updates (Iwasawa & Matsuo, 2021; Karmanov et al., 2024; Zhang et al., 2023). For instance, T3A (Iwasawa & Matsuo, 2021) uses prototypes as classifiers and adaptively reweights them based on test data. AdaNPC (Zhang et al., 2023) mitigates computational overhead and domain forgetting by incorporating source domain data into its prototype framework. TDA (Karmanov et al., 2024) introduces positive and negative caches to curate high-quality test samples from the target domain, enhancing prediction reliability. However, these methods struggle when downstream tasks demand fine-grained understanding or when the test distribution is considerably different from the training distribution.

In contrast, training-required TTA methods adjust specific model components, such as prompts (Shu et al., 2022; Samadh et al., 2023) or batch normalization layers (Wang et al., 2020) to improve performance on target tasks without additional training data. For example, Tent (Wang et al., 2020) minimizes entropy during inference to enhance generalization on distribution-shifted data. Test-time prompt tuning (TPT) (Shu et al., 2022) dynamically refines prompts at test time to bolster zero-shot capabilities. TPT generates multiple augmented views of each test sample and reduces the entropy of the model's output logits across these views, promoting prediction consistency. Building on this, recent advancements like DiffTPT (Feng et al., 2023) utilize diffusion models to create semantically coherent augmentations for entropy minimization, while PromptAlign (Samadh et al., 2023) aligns token statistics (e.g., mean and variance) between test samples and the source distribution to bridge domain gaps. More recently, BoostAdapter (Zhang et al., 2022b) bridges training-required and training-free methods by introducing a lightweight memory-based retrieval mechanism that leverages both historical and instance-aware samples, achieving state-of-the-art performance on TTA of VLMs. ZERO (Farina et al., 2024) is a concurrent episodic TTA method that requires no parameter updates: it augments a test image $N$ times, retains the most confident predictions, and marginalises with softmax temperature set to zero, requiring a single forward pass. While efficient, ZERO operates in the one-sample episodic regime and does not update the model encoder. PseudoAdapter, by contrast, adapts the encoder online via gradient-based LoRA updates over streaming batches, targeting large distribution shifts where feature representations need to be updated. StatA (Zanella et al., 2025) proposes a transductive batch-level algorithm with anchor-based regularisation that processes the entire test set collectively to correct for class distribution shift, whereas PseudoAdapter adapts the encoder online in a streaming fashion without requiring access to the full test batch or any assumptions about class distribution. WATT (Osowiechi et al., 2024) performs full test-time adaptation of CLIP by deriving pseudo-labels from a diverse set of text-prompt templates, updating the model under each template, and consolidating the updates through weight averaging, combined with a text-ensemble strategy at inference. CLIPTTA (Lafon et al., 2026) is a

gradient-based VLM TTA method that replaces entropy minimization with a soft contrastive loss aligned with CLIP's image-text pre-training objective, using a batch-aware design that mitigates pseudo-label drift and class collapse. However, two key limitations remain: reliance on pseudo-label predictions, which are often inaccurate, and using the pre-trained model purely as a frozen feature extractor, which is insufficient for adapting to target domains with significant distribution shifts.

# 3 Method

## 3.1 Preliminaries

In this section, we outline the foundational concepts and notations for TTA. Let's define the target domain distribution as $p_t(x, y)$, representing the joint distribution of input sample $x \in \mathcal{X}$ and their corresponding labels $y \in \mathcal{Y}$, where $\mathcal{Y} = \{1, \ldots, N\}$ denotes a set of $N$ classes. Unlike traditional domain adaptation methods that assume access to source data or labelled samples during adaptation, TTA imposes a stricter and more realistic constraint — it operates solely on unlabeled target data at test time. Specifically, the input samples come as a stream $\{x_i\}_{i=1}^n$, without ground truth labels. The goal is to adapt a pre-trained model to such a stream of samples to maximize the performance.

The pre-trained encoder in our method is CLIP, which is a VLM with image and text encoders, and can perform zero-shot classification. Let $g : \mathcal{X} \to \mathbb{R}^M$ denote the image encoder, mapping inputs to a $M$-dimensional feature space, and $w_c \in \mathbb{R}^M$ represent the text embedding for class $c$, derived from a prompt (e.g., "a photo of a [class]"). Both $g(x)$ and $w_c$ are normalized to unit norm. The logit for class $c$ is computed as: $Z_c = w_c^\top g(x)$, yielding a probability distribution over $N$ classes via softmax: $p(x) = \text{softmax}([Z_1, Z_2, \ldots, Z_N])$. This prompt-based formulation allows CLIP to flexibly generalize to unseen class names and novel domains without additional training. However, under significant distribution shifts, the pre-trained VLM may fail to align with the target distribution of the test data, which requires TTA to improve performance in such scenarios.

In the absence of output labels, most existing TTA methods rely on pseudo-labels to adapt the model. For a test sample $x$, a pseudo-label $\hat{y}$ is assigned by the model prediction,

$$\hat{y}_i = \arg\max_c p_c(x_i). \tag{1}$$

Different TTA methods vary in how they utilize these pseudo-labels, where most SOTA methods such as (Zhang et al., 2022b) focus on linear probing or prototype learning with a frozen encoder. However, under large distribution shifts, the representation captured by a frozen encoder may not effectively reflect the target distribution. Furthermore, the pseudo-labels generated by the pre-trained VLM, especially at the beginning of adaptation, are often noisy. These inaccurate pseudo-labels lead to suboptimal supervision, ultimately degrading adaptation performance.

## 3.2 PseudoAdapter

To address the limitations of existing TTA methods as discussed above, we propose a novel solution called PseudoAdapter. Unlike prior methods that rely on learning simple feature transformations following a frozen pre-trained encoder, PseudoAdapter enables the learning of domain-specific features by fine-tuning the encoder using LoRA (Hu et al., 2022). This approach mitigates the constraints of frozen encoders and enables adaptation to large distribution shifts. Prior works have shown that excessive tuning of model parameters can lead to loss of generalization in pre-trained VLMs (Roy & Etemad, 2024; Roy et al., 2025). Accordingly, to mitigate the risk of overfitting or loss of generalization, PseudoAdapter limits the number of trainable parameters at the fine-tuning stage by incorporating LoRA only into the first $L$ layers of the image encoder of the VLM. For layer $l$ ($\leq L$), with weight matrix $W_l \in \mathbb{R}^{p \times q}$, the LoRA adaptation can be defined as:

$$W_l' = W_l + \Delta W_l, \tag{2}$$

where $\Delta W_l = A_l B_l$, $A_l \in \mathbb{R}^{p \times r}$ and $B_l \in \mathbb{R}^{r \times q}$ are learnable matrices, and $r \ll \min(p, q)$ is the rank of LoRA.

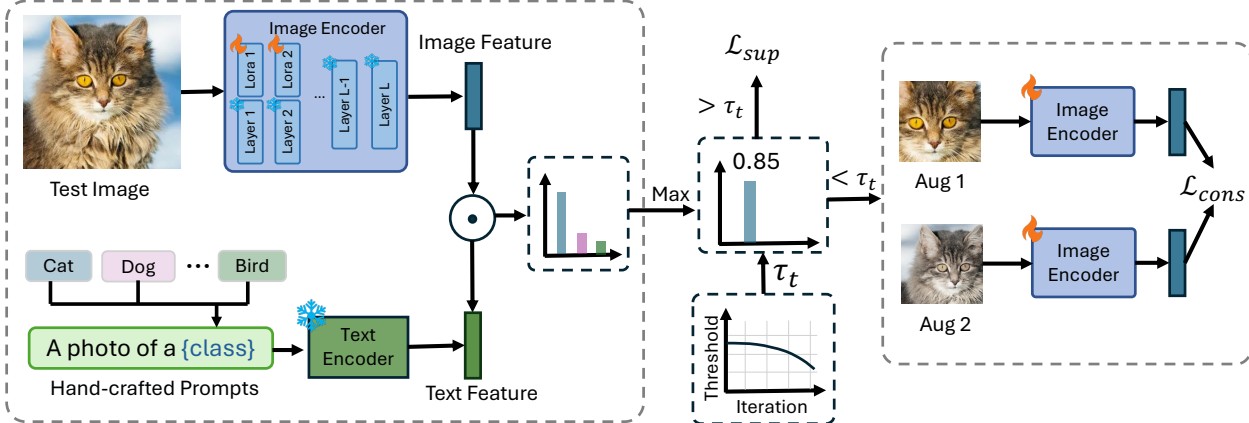

Figure 3: Overview of PseudoAdapter. We fine-tune the model by incorporating LoRA modules into the pre-trained encoder. These modules are added only to the early layers of the encoder. The model is trained with a supervised loss using high-confidence pseudo-labels and an unsupervised loss for the remaining samples.

To learn the target distribution, we propose a pseudo-consistency learning approach that, instead of relying solely on pseudo-label predictions and utilizing high-confidence samples (Zhang et al., 2022b), leverages the entire test set to effectively adapt to the target domain, while mitigating the impact of noisy pseudo-labels. To achieve this, we combine supervised learning on high-confidence samples with unsupervised learning on the remaining samples. Given an image $x_i$, we first generate the pseudo-label using the VLM as $\hat{y}_i = \arg\max_c p_c(x_i)$, where $p_c(x_i)$ denotes the softmax probability of class $c$. We consider this prediction as a high-confidence pseudo-label if its prediction confidence satisfies $p_{\hat{y}_i}(x_i) \geq \tau$, where $\tau$ is a confidence threshold. This confidence-based filtering of pseudo-labels follows FixMatch (Sohn et al., 2020); unlike FixMatch's fixed threshold, we adapt it with a dynamic schedule (introduced below) suited to the TTA setting. For these samples, we treat pseudo-labels as ground truth for supervised learning, and train the model with a cross-entropy loss as:

$$\mathcal{L}_{\text{sup}} = \ell_{\text{CE}}(p(x_i), \hat{y}_i). \tag{3}$$

However, at the early stages of adaptation, pseudo-label predictions are often noisy due to miscalibration, leading to incorrect predictions with high confidence. To address this, we introduce a confidence calibration module that dynamically adjusts the confidence threshold for pseudo-labels. At the start of adaptation (time step $t = 0$), PseudoAdapter sets a high threshold ($\tau_0$) to ensure only very high-confidence predictions are considered as pseudo-labels. Over $T$ adaptation steps, it lowers $\tau$ using a linear schedule to gradually allow for more pseudo-labels to contribute towards supervised learning. The threshold at time step $t$ can be defined as:

$$\tau_t = \tau_0 - (\tau_0 - \tau_{\min}) \cdot \frac{t}{T}, \tag{4}$$

where $\tau_{\min}$ is the final threshold.

On the other hand, to facilitate learning from the remaining (low-confidence) samples, we incorporate a contrastive loss that learns representations in an unsupervised manner. Specifically, the model learns representations by pulling together positive samples while pushing apart negative samples in the embedding space. For a sample $x_i$, we apply data augmentations (e.g., random crops, horizontal flips, colour jitter) to form the positive set $x_i^{(1)}, x_i^{(2)}, \ldots, x_i^{(K)}$ consisting of $K$ samples. Inspired by MoCo (He et al., 2020), we form the negatives by storing samples from previous iterations in a memory bank, $\mathcal{M} = \{m_1, m_2, \ldots, m_M\} \subset \mathbb{R}^C$, where $M$ is the size of the memory bank. For a sample $x_i$, the contrastive loss is defined as:

$$\mathcal{L}_{\text{con}} = -\frac{1}{K} \sum_{k=1}^{K} \log \frac{\sum_{l=1, l \neq k}^{K} s_{kl}}{\sum_{l=1, l \neq k}^{K} s_{kl} + \sum_{m_j \in \mathcal{M}} t_{kj}}. \tag{5}$$

Table 1: Comparison to existing methods on the OOD benchmark. We report top-1 accuracy, and the average is computed as the mean accuracy across all four OOD datasets.

| Method | Imagenet-V2 | Imagenet-S | Imagenet-A | Imagenet-R | Average |
|---|---|---|---|---|---|
| CLIP (Radford et al., 2021) | 60.86 | 46.09 | 47.87 | 73.98 | 57.20 |
| CLIP+TPT (Shu et al., 2022) | 64.35 | 47.94 | 54.77 | 77.06 | 60.81 |
| CoOp (Zhou et al., 2022b) | 64.20 | 47.99 | 49.71 | 75.21 | 59.28 |
| CoOp+TPT (Shu et al., 2022) | 66.83 | 49.29 | 57.95 | 77.27 | 62.84 |
| Co-CoOp (Zhou et al., 2022a) | 64.07 | 48.75 | 50.63 | 76.18 | 59.91 |
| Co-CoOp+TPT (Shu et al., 2022) | 64.85 | 48.27 | 58.47 | 78.65 | 62.61 |
| Maple (Khattak et al., 2023) | 64.07 | 49.15 | 50.90 | 76.98 | 60.28 |
| Maple + TPT (Shu et al., 2022) | 64.87 | 48.16 | 58.08 | 78.12 | 62.31 |
| PromptAlign (Samadh et al., 2023) | 65.29 | 50.23 | 59.37 | 79.33 | 63.55 |
| DiffTPT (Feng et al., 2023) | 65.10 | 46.80 | 55.68 | 75.00 | 60.52 |
| TDA (Karmanov et al., 2024) | 64.67 | 50.54 | 60.11 | 80.24 | 63.89 |
| TTL (Imam et al., 2025) | 64.55 | 48.62 | 60.51 | 77.54 | 62.80 |
| BoostAdapter (Zhang et al., 2022b) | 65.51 | 51.28 | 64.53 | 80.95 | 65.57 |
| **PseudoAdapter** | 66.74 | 52.78 | 68.42 | 82.19 | 67.53 |
| **PseudoAdapter+** | **69.10** | **57.73** | **75.08** | **87.71** | **72.41** |
| Δ | ↑ 3.59 | ↑ 6.45 | ↑ 10.55 | ↑ 6.76 | ↑ 6.84 |

where $s_{kl} = \exp\left(g(x_i^{(k)})^\top g(x_i^{(l)})/\gamma\right)$, and $t_{kj} = \exp\left(g(x_i^{(k)})^\top m_j/\gamma\right)$ ,, $\gamma$ is the temperature hyperparameter for the contrastive loss. Finally, the overall loss of PseudoAdapter can be represented as:

$$\mathcal{L} = \mathcal{L}_{\text{sup}} + \lambda \mathcal{L}_{\text{con}}, \tag{6}$$

where $\lambda$ is a weighting factor.

### 3.3 PseudoAdapter+

To further enhance the adaptation of the pre-trained encoder, we introduce PseudoAdapter+, a teacher-guided adaptation strategy that selectively gathers teacher supervision to improve test-time adaptation on the target domain. Here, the teacher encoder is a larger pre-trained model with stronger generalization capabilities than the student (trainable encoder in PseudoAdapter). To maintain computational efficiency while utilizing teacher supervision, we selectively query the teacher only for samples where the student model's confidence is lower than a threshold $\tau_t$: $p_{\hat{y}_i}(x_i) < \tau_t$. PseudoAdapter+ ensures minimal teacher supervision by setting $\tau_t$ to a small value. For a sample, $x_i$ with $p_{\hat{y}_i}(x_i) < \tau_t$, we predict the pseudo-label with the teacher encoder as $y_i^{\text{teacher}} = \arg\max_c p_c^{\text{teacher}}(x_i)$, where $p^{\text{teacher}}(x_i)$ is the teacher's softmax probability. Then we train the student with the updated pseudo-label as:

$$\mathcal{L}_{\text{teacher}} = \frac{1}{|\mathcal{T}_t|} \sum_{x_i \in \mathcal{T}_t} \ell_{\text{CE}}(p(x_i), y_i^{\text{teacher}}). \tag{7}$$

Overall, we treat PseudoAdapter+ as a variant of our proposed method and present final results for both the base PseudoAdapter (without teacher guidance) and PseudoAdapter+ in our experiments.

Table 2: Full results on the cross-domain benchmark. We report top-1 accuracy, and the average is calculated as the mean accuracy across all ten datasets.

| Method | Caltech | Pets | Cars | Flowers | Food101 | Aircraft | SUN397 | DTD | EuroSAT | UCF101 | *Average* |
|---|---|---|---|---|---|---|---|---|---|---|---|
| CLIP (Radford et al., 2021) | 93.35 | 88.25 | 65.48 | 67.44 | 83.65 | 23.67 | 62.59 | 44.27 | 42.01 | 65.13 | 63.58 |
| CLIP+TPT (Shu et al., 2022) | 94.16 | 87.79 | 66.87 | 68.98 | 84.67 | 24.78 | 65.50 | 47.75 | 42.44 | 68.04 | 65.10 |
| CoOp (Zhou et al., 2022b) | 93.70 | 89.14 | 64.51 | 68.71 | 85.30 | 18.47 | 64.15 | 41.92 | 46.39 | 66.55 | 63.88 |
| CoCoOp (Zhou et al., 2022a) | 93.79 | 90.46 | 64.90 | 70.85 | 83.97 | 22.29 | 66.89 | 45.45 | 39.23 | 68.44 | 64.63 |
| MaPLe (Khattak et al., 2023) | 93.53 | 90.49 | 65.57 | 72.23 | 86.20 | 24.74 | 67.01 | 46.49 | 48.06 | 68.69 | 66.30 |
| MaPLe+TPT (Shu et al., 2022) | 93.59 | 90.72 | 66.50 | 72.37 | 86.64 | 24.70 | 67.54 | 45.87 | 47.80 | 69.19 | 66.50 |
| DiffTPT (Feng et al., 2023) | 92.49 | 88.22 | 67.01 | 70.10 | 87.23 | 25.60 | 65.74 | 47.00 | 43.13 | 62.67 | 65.47 |
| PromptAlign (Samadh et al., 2023) | 94.01 | 90.76 | 68.50 | 72.39 | 86.65 | 24.80 | 67.54 | 47.24 | 47.86 | 69.47 | 66.92 |
| TTL (Imam et al., 2025) | 93.63 | 88.72 | 67.96 | 70.48 | 85.05 | 23.82 | 66.32 | 46.69 | 42.02 | 69.20 | 65.39 |
| TDA (Karmanov et al., 2024) | 94.24 | 88.63 | 67.28 | 71.42 | 86.14 | 23.91 | 67.62 | 47.40 | 58.00 | 70.66 | 67.53 |
| BoostAdapter (Zhang et al., 2022b) | 94.77 | 89.51 | 69.30 | 71.66 | 87.17 | 27.45 | 68.09 | 45.69 | 61.22 | 71.93 | 68.68 |
| **PseudoAdapter** | 95.01 | 91.10 | 71.39 | 72.59 | 88.95 | 29.41 | 68.64 | 46.27 | 61.98 | 72.98 | 69.83 |
| **PseudoAdapter+** | **95.29** | **93.62** | **73.45** | **77.26** | **90.25** | **31.45** | **70.14** | **50.36** | **63.45** | **74.03** | **71.93** |
| Δ | ↑ 0.28 | ↑ 4.11 | ↑ 4.15 | ↑ 5.6 | ↑ 3.08 | ↑ 4.0 | ↑ 2.05 | ↑ 4.67 | ↑ 2.23 | ↑ 2.10 | ↑ 3.25 |

Table 3: Comparisons with baselines on ImageNet-C at severity level 5.

| | Noise | | | Blur | | | | Weather | | | | Digital | | | | |
|---|---|---|---|---|---|---|---|---|---|---|---|---|---|---|---|---|
| | Gauss. | Shot | Impul. | Defoc. | Glass | Motion | Zoom | Snow | Frost | Fog | Brit. | Contr. | Elastic | Pixel | JPEG | Avg. |
| CLIP | 15.15 | 16.28 | 15.26 | 25.83 | 16.87 | 26.34 | 24.43 | 34.56 | 33.01 | 39.10 | 57.78 | 18.45 | 14.71 | 35.62 | 35.81 | 27.28 |
| TDA | 17.50 | 18.59 | 18.12 | 59.12 | 19.02 | 28.25 | 26.24 | 37.30 | 35.30 | 41.57 | 59.04 | 21.06 | 17.61 | 37.78 | 37.26 | 31.58 |
| BoostAdapter | 17.53 | 18.89 | 18.39 | 59.70 | 19.07 | 28.62 | 27.33 | 38.21 | 36.13 | 42.31 | 59.63 | 21.22 | 18.23 | 39.25 | 38.07 | 32.17 |
| **PseudoAdapter** | 18.32 | 19.23 | 18.92 | 60.11 | 19.98 | 29.61 | 28.01 | 39.11 | 36.90 | 43.78 | 60.45 | 22.23 | 19.03 | 38.39 | 38.87 | 32.86 |
| **PseudoAdapter+** | **20.34** | **23.04** | **20.44** | **63.12** | **22.02** | **32.23** | **30.47** | **42.67** | **38.28** | **45.08** | **62.48** | **25.05** | **21.75** | **40.32** | **39.73** | **35.13** |
| Δ | ↑ 2.81 | ↑ 4.15 | ↑ 2.05 | ↑ 3.42 | ↑ 2.95 | ↑ 3.61 | ↑ 3.14 | ↑ 4.46 | ↑ 2.15 | ↑ 2.77 | ↑ 2.85 | ↑ 3.83 | ↑ 3.52 | ↑ 1.07 | ↑ 1.66 | ↑ 2.96 |

## 4 Experiments

### 4.1 Datasets and Implementation Details

Following the evaluation protocol of prior works (Shu et al., 2022; Zhang et al., 2022b; Karmanov et al., 2024), we evaluate our method on two distinct benchmarks: an out-of-distribution (OOD) benchmark and a cross-domain benchmark. The OOD benchmark tests robustness to natural distribution shifts using four ImageNet (Deng et al., 2009) variants: ImageNetV2 (Recht et al., 2019), ImageNet-S (Wang et al., 2019), ImageNet-A (Hendrycks et al., 2021b), and ImageNet-R (Hendrycks et al., 2021a). For the cross-domain benchmark, we evaluate performance across ten diverse datasets: Aircraft (Maji et al., 2013), Caltech101 (Fei-Fei et al., 2004), Cars (Krause et al., 2013), DTD (Cimpoi et al., 2014), EuroSAT (Helber et al., 2019), Flower102 (Nilsback & Zisserman, 2008), Food101 (Bossard et al., 2014), Pets (Parkhi et al., 2012), SUN397 (Xiao et al., 2010), and UCF101 (Soomro et al., 2012). We adopt the data splits from (Zhou et al., 2022b) and report the accuracy.

All the experiments are conducted with a pre-trained CLIP-B/16 as the base model and a pre-trained CLIP-L/14 as the teacher, where applicable. To fine-tune the model, we incorporate low-rank adapters of rank 16, and train the model with Adam optimizer and a learning rate of 5e-5. We follow BoostAdapter (Zhang

Table 4: Ablation analysis of different components of our method.

(a) **Component Ablation**

| Configuration | Acc. (%) |
|---|---|
| PseudoAdapter | $68.42 \pm 0.15$ |
| w/o LoRA | $64.34 \pm 0.19$ |
| w/o Unsup. loss | $67.06 \pm 0.18$ |
| w/o Confidence Cal. | $67.62 \pm 0.21$ |
| w/ last-$L$ LoRA | $65.27 \pm 0.17$ |
| w/ norm. layers only | $65.73 \pm 0.20$ |

(b) **Unsupervised Loss**

| Loss | Acc. |
|---|---|
| None | 67.06 |
| KL-div | 66.48 |
| VICReg | 68.03 |
| Contrastive | **68.42** |

(c) **Dynamic Thresholding**

| Method | Acc. |
|---|---|
| Static | 67.62 |
| Linear | **68.42** |
| Exponential | 68.15 |
| Cosine | 68.20 |

Table 5: Ablation on augmentation strategy for the contrastive loss (ImageNet-A).

| Augmentation | Acc. (%) |
|---|---|
| Horizontal flip only | 67.6 |
| Random crop only | 67.9 |
| Color jitter only | 67.5 |
| Crop + flip | 68.1 |
| Crop + flip + jitter (full, default) | **68.42** |

et al., 2022b) for all other default hyperparameters and experimental setups. All the parameters specific to our method are discussed in the sensitivity study in Section 4.3. All experiments are conducted on an NVIDIA V100 GPU with 32GB of memory.

## 4.2 Main Results

**OOD Generalization.** We present the performance of PseudoAdapter on the OOD evaluation benchmarks in Table 1. As shown, PseudoAdapter consistently outperforms the existing SOTA method, BoostAdapter. On average, PseudoAdapter achieves a 1.96% improvement, with gains of 1.23%, 1.50%, 3.89%, and 1.24% on ImageNet-V2, ImageNet-S, ImageNet-A, and ImageNet-R, respectively. Furthermore, PseudoAdapter+ yields additional performance gains, achieving an average improvement of 6.84% over BoostAdapter. It improves accuracy by 3.59%, 6.45%, 10.55%, and 6.76% on the respective datasets. These results demonstrate the effectiveness of our method in addressing domain shifts and its strong generalization across challenging OOD benchmarks.

**Cross-domain Transfer.** Table 2 presents results on the cross-domain benchmark. As we find from this table, PseudoAdapter demonstrates strong performance, achieving an average improvement of 1.15% over BoostAdapter, with gains of up to 1.96% on individual datasets. Notably, it outperforms BoostAdapter on all ten datasets. Furthermore, PseudoAdapter+ provides additional improvements over both PseudoAdapter and BoostAdapter across the datasets by achieving a 3.25% average improvement over the previous SOTA and with gains of up to 4.67% on individual datasets.

**Generalization on Corruption Datasets.** To further evaluate the performance of PseudoAdapter under distribution shifts, we report results on the ImageNet-C dataset at the highest severity level (level 5). As shown in Table 3, PseudoAdapter consistently improves performance across all 15 corruption types, with an average improvement of 0.69%. Furthermore, PseudoAdapter+ provides an additional boost over existing methods, outperforming BoostAdapter by an average of 2.96%.

## 4.3 Ablation and Sensitivity Studies

To understand the contribution of each component in our method, we conduct a series of ablation and sensitivity studies. All experiments are performed on a OOD dataset (ImageNet-A). Specifically, we ablate

Table 6: Time complexity of different methods on ImageNet-A on an NVIDIA V100 GPU.

| Method | Inf. Speed (FPS) | Performance |
|---|---|---|
| CLIP (baseline) | 15.42 | 47.87 |
| TDA | 15.40 | 60.11 |
| BoostAdapter | 15.21 | 64.53 |
| PseudoAdapter | 15.11 | 68.42 |
| PseudoAdapter+ | 13.45 | 75.08 |

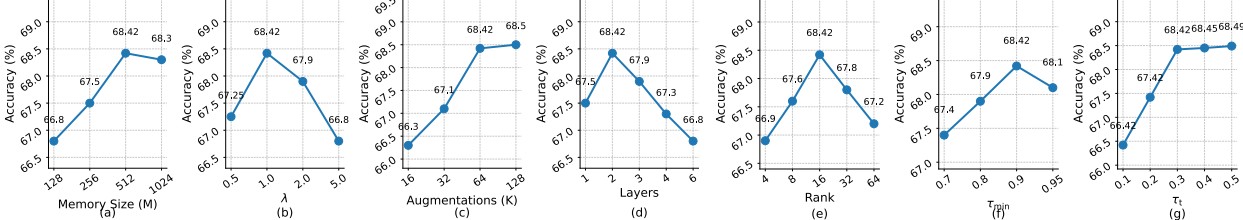

Figure 4: Hyper-parameter study on (a) memory bank size, (b) unsupervised loss weight, (c) number of augmentations, (d) number of LoRA layers, (e) rank of LoRA layers, (f) adaptive threshold, and (g) teacher guidance threshold.

each component of PseudoAdapter, explore variations in the unsupervised loss functions, explored different decay methods for the dynamic threshold, and perform sensitivity studies on the memory bank size, unsupervised loss factor, number of augmentations, number of LoRA layers, rank of LoRA, threshold for confidence calibration $\tau_{min}$, and teacher guidance threshold $\tau_t$. The results are summarized in Table 4 and Figure 4.

In Table 4a, we present an ablation study on the main components of PseudoAdapter and observe that removing LoRA leads to a significant drop of 4.08%, which highlights that fine-tuning the encoder is crucial for effective test-time adaptation. Similarly, removing the unsupervised loss reduces accuracy by 1.36%, indicating the importance of learning from all samples for robust adaptation. Excluding the confidence calibration module also lowers accuracy by 0.80%, emphasizing its role in mitigating the impact of noisy pseudo-labels during adaptation. In Table 4b, we compare different unsupervised losses for learning from low-confidence samples and observe that contrastive loss achieves the highest accuracy of 68.42%, followed by VICReg (Bardes et al., 2021) with 68.03%, while KL-divergence (consistency loss implemented as the KL divergence $\mathrm{KL}(p(x_i^{(1)}) \| p(x_i^{(2)}))$) between predictions on two augmented views of the same sample) performs the worst with an accuracy of 66.48%. Next, we evaluate different strategies for adjusting the threshold in our dynamic thresholding approach, as shown in Table 4c by comparing static, linear, exponential, and cosine schedules. Among these, the linear schedule for adjusting $\tau_t$ achieves the highest accuracy of 68.42%, followed by exponential and cosine schedules.

Next, we present a study on the augmentation types in Table 5. The results show that the performance difference across augmentation strategies is modest (within 1%), suggesting the contrastive loss is reasonably robust to the specific augmentation choice as long as meaningful view diversity is provided. Random cropping contributes the most among individual augmentations, as spatially diverse crops generate views with varied local contexts that are most informative for contrastive learning. The more critical design choice is the number of augmented views (Figure 4c), where reducing to 16 or 32 views causes a more pronounced drop, indicating that view diversity matters more than the specific augmentation type.

In Figure 4a, we examine the impact of memory bank size, $M$ and observe that a size of 512 achieves the highest accuracy, while smaller sizes (256 and 128) result in performance drops due to insufficient negative samples for contrastive learning. Interestingly a larger size of 1024 slightly reduces accuracy, possibly due to increased noise or redundancy in the memory bank, suggesting that 512 strikes an optimal balance given our current architecture. Study on the unsupervised loss weight $\lambda$ in Figure 4b shows that $\lambda = 1.0$ yields the best accuracy while lower ($\lambda = 0.5$) or higher weights ($\lambda = 2.0, 5.0$) degrade performance, indicating that an equal

weighting of supervised and unsupervised losses optimally balances learning in supervised and unsupervised settings. Study on the number of augmentations in Figure 4c shows that using 128 augmented views per sample yields the highest accuracy of 68.50%. However, using 64 views achieves a comparable performance of 68.42%. Since existing methods such as BoostAdapter (Zhang et al., 2022b) use 64 augmentations, we adopt the same setting by default to maintain comparable computational cost. In contrast, using fewer views (16 and 32) results in lower accuracies, indicating that limited augmentation hinders robust contrastive learning. In Figure 4d, we evaluate the performance for fine-tuning different number of LoRA layers ($L$), ranging from 1 to 6 showning that the peak accuracy is achieved with $L = 2$ for effectively balancing the test time adaptation with preserving generalization. Increasing the number of layers to 3, 4, or 6 leads to a gradual decline in accuracy likely due to overfitting as more parameters are tuned. In Figure 4e, we examine the LoRA rank ($r$) with values of 4, 8, 16, 32, and 64 where the highest accuracy is observed at $r = 16$. Lower ranks ($r = 4, 8$) yield reduced performances reflecting insufficient capacity for learning, while higher ranks ($r = 32, 64$) slightly decrease performance due to potential overfitting. In Figure 4f, we analyze the final pseudo-label confidence threshold ($\tau_{\min}$) in the dynamic thresholding schedule, with values of 0.7, 0.8, 0.9, and 0.95. The best performance occurs at $\tau_{\min} = 0.9$, indicating that a relatively high confidence threshold at the end of adaptation ensures reliable supervision from pseudo-labels. Finally in Figure 4g, we investigate the teacher guidance threshold for querying the teacher model for pseudo-labels, with values from 0.1 to 0.5 where we observe that accuracy improves sharply from 66.42% at 0.1 to 68.42% at 0.3, reflecting the benefit of selective teacher supervision for low-confidence samples. Beyond 0.3, performance increases with marginal gains. Since teacher supervision is computationally expensive, we set this threshold to 0.3.

### 4.4   Time Complexity

In Table 6, we present a time complexity analysis of our proposed solution compared to existing SOTA methods, TDA and BoostAdapter. All experiments are conducted on an NVIDIA V100 GPU.

We report inference speed in frames per second (FPS) alongside model performance. As shown, PseudoAdapter achieves a comparable average FPS to TDA and BoostAdapter, while outperforming both methods by significant margins. PseudoAdapter+, although slightly slower due to selective use of a teacher encoder for certain samples, still maintains a competitive speed of 13.45 FPS compared to 15.21 for BoostAdapter, while delivering a substantial performance gain vs. the SOTA BoostAdapter (75.08% vs. 64.53%).

### 4.5   Qualitative Results

Here we present qualitative examples from the ImageNet-S dataset, comparing the top-2 predictions of pre-trained CLIP, previous SOTA BoostAdapter, and our methods PseudoAdapter and PseudoAdapter+. In particular, we observe that the true class of the top image is 'Tabby cat', but the pre-trained CLIP predicts it as 'Persian cat'. The feature transformation approach in BoostAdapter changes the probability of the predictions, but still generates incorrect ones. On the other hand, PseudoAdapter and PseudoAdapter+ generate the correct predictions, with PseudoAdapter+ having a higher confidence for the correct prediction. A similar trend can be observed for the other example depicted in the figure.

## 5   Conclusion

In this work, we introduced PseudoAdapter, a novel test-time adaptation framework for vision-language models that addresses key limitations of existing approaches. Unlike prior methods that rely on frozen feature extractors or noisy pseudo-labels, PseudoAdapter adapts the pre-trained encoder using low-rank adapters inserted in early layers, enabling domain-specific feature learning while preserving generalization. To robustly leverage the entire target domain, we proposed a hybrid learning strategy that combines confidence-calibrated pseudo-label supervision with unsupervised consistency learning, ensuring effective adaptation without overfitting. Furthermore, we extended our approach with PseudoAdapter+, a teacher-guided variant that selectively incorporates supervision for low-confidence samples, improving performance with minimal computational overhead. Extensive experiments on OOD and cross-domain benchmarks demonstrate the superiority of our approach over SOTA methods, both in accuracy and robustness.

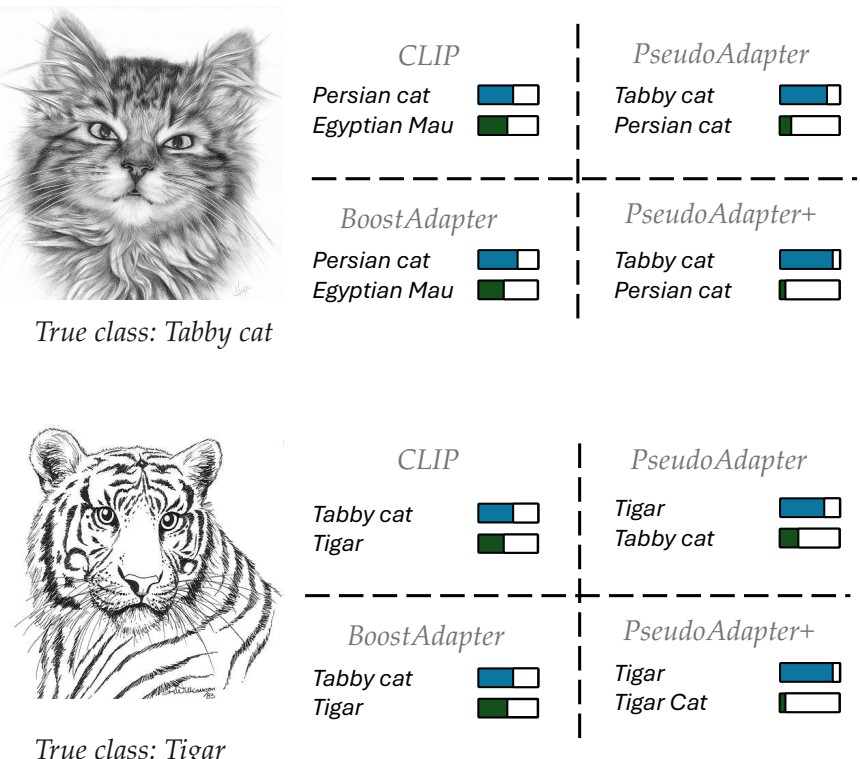

Figure 5: Qualitative examples comparing the predictions of CLIP, BoostAdapter, PseudoAdapter, and PseudoAdapter+.

## Broader Impact

PseudoAdapter improves the robustness of vision-language models to distribution shift, which can benefit deployed systems in safety-critical domains (e.g., medical imaging, autonomous driving) where training and test distributions frequently differ. A potential negative impact is that test-time adaptation could be exploited to adapt a model to adversarial or harmful target distributions without human oversight. We encourage practitioners to apply appropriate monitoring when deploying TTA methods in high-stakes applications.

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
