# OpenReview forum: "Test-Time Adaptation of Vision-Language Models with Low-Rank Pseudo-Consistency"
_TMLR — Accepted by TMLR_

### Review · Reviewer_xUDR · 2026-02-17

**Summary Of Contributions:**

This paper proposes a test-time adaptation (TTA) method called PseudoAdapter for zero-shot classification with CLIP. PseudoAdapter incorporates contrastive learning and supervised learning with high-confidence pseudo labels to mitigate noise from pseudo labels and leverage low-confidence samples. LoRA added to early layers is optimized instead of CLIP parameters to avoid overfitting. Additionally, PseudoAdapter+, a variant of PseudoAdapter, is proposed. PseudoAdapter+ uses a teacher model larger than the adapted model to generate high-quality pseudo labels for low-confidence samples. Experimental results show that PseudoAdapter outperformed CLIP TTA baselines on various domains.

**Additional Comments:**

- The confidence scale can differ among domains or the number of classes. For example, predictions can be very uncertain under image corruption compared to other domain shifts or when the number of classes increases.  Can the hyperparameter setting for the thresholding generalize across the domains or the number of classes?
- How many times was the TTA run for each setting? Especially for the ablation, displaying standard deviations would be helpful to see a significant difference.

**Audience:**

Yes

**Audience Explanation:**

The paper is well-written and easy to follow, and the procedure of the proposed method is easy to understand.

**Broader Impact Concerns:**

The broader impact section should be explicitly added.

**Claims And Evidence:**

Yes

**Claims Explanation:**

The claims are experimentally validated.

**Requested Changes:**

## Major
- The proposed method lacks novelty since it mostly consists of existing components, i.e., pseudo labels, contrastive learning, and teacher supervision. Explaining how they complementarily work would be convincing. It is also unclear why the target of PseudoAdapter is limited to VLMs while it can be applied to general image classification models. More discussion on existing TTA methods and related components not limited to VLMs, such as [a-e], would be beneficial.
- The scheduling of the confidence threshold in Eq. (4) relies on the total number of steps $T$, while $T$ may be unknown in the real-world setting. How should $T$ be selected in practice?
- PseudoAdapter+ assumes access to a larger teacher model during TTA. Do we need TTA for the student model in this setting? For low-confidence samples, the teacher outputs can be directly used for inference, i.e., cancade inference [b].
- For Tabs. 1-3, reporting the accuracy of the teacher CLIP would be helpful to see the upper bound performance.
- A more inference speed analysis, especially for PseudoAdapter+, would be interesting. There should be a trade-off between the threshold $\tau_t$ and the throughput. Plotting the trade-off would be helpful for practitioners.


[a] Chen et al., Contrastive Test-Time Adaptation, CVPR2022.
[b] Enomoto and Eda, Learning to Cascade: Confidence Calibration for Improving the Accuracy and Computational Cost of Cascade Inference Systems, AAAI2021.
[c] Yuan et al., Robust Test-Time Adaptation in Dynamic Scenarios, CVPR2023.
[d] Wang et al., Continual Test-Time Domain Adaptation, CVPR2022.
[e] Ma, Improved Self-Training for Test-Time Adaptation, CVPR2024.

## Minor
- `\citep` and `\citet` commands should be used properly for citations for readability.
- $\gamma$ in Eq. (5) is undefined (temperature?).
- A more description of the KL divergence in Tab. 4 (b) would be helpful.

---

> ### Author Response · Authors · 2026-03-19
>
> > Scope limited to VLMs
>
> The fundamental difference between a VLM and a vision-only model mainly lies in the zero-shot capabilities of VLMs. We do not introduce the first TTA approach for VLMs, as this area is already well established in the literature, and methods developed for VLMs are not directly transferable to vision-only models. Similarly, the evaluation protocols differ due to inherent capability gaps. For example, vision-only models cannot perform cross-dataset evaluation with varying class distributions, a setting in which VLMs are commonly evaluated. Therefore, results in [a–e] are not directly comparable.
>
> > Limited novelty: combination of existing components
>
> While individual components have conceptual roots in existing literature, no prior work has applied any of these components — individually or jointly — to the problem of test-time adaptation of VLMs. Pseudo-labels, contrastive learning, and teacher supervision each originate from different domains (semi-supervised learning, self-supervised pre-training, and offline knowledge distillation, respectively), and transplanting these ideas into the online, unlabelled, streaming TTA setting is non-trivial and requires specific adaptations. More importantly, each component is designed to address a specific, identifiable failure mode in existing VLM TTA methods: (1) The frozen encoder in existing TTA methods cannot adapt low-level features under large domain shifts; early-layer LoRA directly addresses this by enabling encoder-level adaptation while preserving deep semantic representations. (2) VLMs are demonstrably miscalibrated at the start of TTA (Figure 2), producing high-confidence but incorrect predictions; a static FixMatch-style threshold would admit corrupted pseudo-labels early in adaptation, so the dynamic threshold is specifically designed to handle this VLM-specific calibration problem. (3) Discarding low-confidence samples (as all existing VLM TTA methods do) wastes the majority of the test stream; the contrastive loss recovers their signal without requiring labels. (4) For the hardest samples, selective teacher supervision provides reliable pseudo-labels while keeping overhead below 5\%, unlike standard offline distillation, which processes all samples equally. The contribution is therefore the identification of these failure modes and the demonstration that this particular composition resolves them.
>
> ---
>
> > T is unknown in practice for streaming settings
>
> In our formulation, $T$ controls the rate at which the confidence threshold decays. Its role is purely to pace the relaxation of the threshold: a larger $T$ means a slower and more conservative decay, while a smaller $T$ means faster relaxation. In practice, we simply set $T$ to the length of the test stream, which is known in standard benchmark evaluation. For open-ended streaming deployments where stream length is unknown, setting $T$ to a conservatively large fixed value ensures the threshold decays slowly and safely, spending more time in the high-threshold regime where pseudo-labels are most reliable. To test this, we present an experiment with a fixed value of $T$ on the ImageNet-A dataset, which shows stable performance for reasonably large $T$ (e.g. 1000).
>
> | Setting | Acc |
> |-------|-------|
> | Default | 68.42|
> | w/ T=500 | 69.35 |
> | w/ T=1000 | 68.40
> | w/ T=2000 | 68.39
>
> ---
>
> > PseudoAdapter+ assumes access to a larger teacher model during TTA. Do we need TTA for the student model in this setting? For low-confidence samples, the teacher outputs can be directly used for inference, i.e., **Cancade inference [b]**.
>
> Yes, we do. Cascade inference routes low-confidence samples directly to the teacher at inference time without updating the student. Unlike Cancade inference [b], PsudoAdapter tunes the student model to learn from the current sample (also utilizing teacher signal when available) to perform better later on.
>
> ---

---

> > ### Author Response · Authors · 2026-03-19
> >
> > > For Tables 1-3, reporting the accuracy of the teacher CLIP would be helpful to see the upper bound performance.
> >
> > The zero-shot performance of CLIP-L/14 is not an upper bound for our method, as the teacher model is not fine-tuned on the downstream datasets, whereas the student adapts during test time. Therefore, comparing against the teacher’s zero-shot performance does not necessarily provide meaningful insight.
> > Additionally, all methods in Tables 1–3 report results using CLIP-B/16, making comparisons to CLIP-L/14 less fair due to differences in model capacity. Nevertheless, to provide context, we include CLIP-L/14 results in the table below. These show that PseudoAdapter+ achieves 72.41\% on average with the base model (CLIP-B/16: 57.20\% zero-shot accuracy), outperforming the zero-shot accuracy of the larger teacher model (CLIP-L/14: 74.76\%).
> >
> > |Model | Imagenet-V2 | Imagenet-S | Imagenet-A | Imagenet-R | Average|
> > |------|------------|------------|------------|-------------|--------|
> > | CLIP-L/14 | 70.8 | 64.7 | 59.2 | 89.3 | 71.0 |
> >
> > ---
> >
> > > Speed–threshold trade-off for PseudoAdapter+Speed–threshold trade-off for PseudoAdapter+
> >
> > The speed–accuracy trade-off for PseudoAdapter+ is directly governed by the teacher guidance threshold τ_t, which controls the fraction of samples routed to the teacher. This trade-off is intuitive and monotonic: a higher τ_t queries the teacher more frequently, improving accuracy at the cost of inference speed. We deliberately set τ_t = 0.3 to keep teacher consultations minimal — at this threshold, fewer than 30\% of samples require teacher supervision, resulting in less than 5\% computational overhead over PseudoAdapter (Table 5).
> >
> >
> > ---
> >
> > > Confidence threshold generalization across domains and number of classes
> >
> > While absolute softmax probability can influenced by the number of classes, in practice, our default threshold transfers well across all evaluated benchmarks, which span a wide range of class counts (up to 1000 classes) across the ten cross-domain datasets in Table 2, without any per-dataset tuning. This suggests that the threshold is robust in practice, likely because CLIP's strong pre-training produces consistently well-separated confidence distributions across tasks regardless of class count.
> >
> > ---
> >
> > > Standard deviations missing from ablation tables
> >
> > All the experiments in the paper are conducted over 3 runs. We have revised Table 4 to include the standard deviation. Standard deviations are ≤ 0.21\% across all configurations, confirming that the differences in Table 4 are statistically meaningful. We update all ablation sub-tables:
> >
> >
> >
> > ---
> >
> > > Minor comments
> >
> > All addressed in the revised paper
> >
> > ---
> >
> > > Broader Impact section:
> >
> > We have added a separate broader impact section:
> >
> > ```
> > PseudoAdapter improves the robustness of vision-language models to distribution shift, which can benefit deployed systems in safety-critical domains (e.g., medical imaging, autonomous driving) where training and test distributions frequently differ. A potential negative impact is that test-time adaptation could be exploited to adapt a model to adversarial or harmful target distributions without human oversight. We encourage practitioners to apply appropriate monitoring when deploying TTA methods in high-stakes applications.
> > ```

---

### Review · Reviewer_nMny · 2026-02-24

**Summary Of Contributions:**

The authors propose PseudoAdapter, a framework for adapting Vision-Language Models (VLMs) at test time. To move away from relying purely on frozen encoders and noisy pseudo-labels, the method introduces three main pieces:

- Early-layer LoRA: Injecting Low-Rank Adapters into the early layers of the VLM's image encoder to learn domain-specific features without overfitting.

- Pseudo-Consistency Learning: A hybrid loss that applies supervised learning to high-confidence pseudo-labels (using a dynamically adjusting threshold) and unsupervised contrastive learning to the remaining low-confidence samples across augmented views.

- PseudoAdapter+: An extension that queries a larger, pre-trained teacher model (CLIP-L/14) to generate pseudo-labels for samples where the student model (CLIP-B/16) has low confidence.

**Audience:**

Yes

**Audience Explanation:**

The adaptation of large-scale foundation models is a highly active research area. TMLR readers working on Test-Time Adaptation, parameter-efficient fine-tuning (like LoRA), and semi-supervised learning would be interested in the empirical results of this paper, particularly the analysis of adapting early layers versus later layers in the VLM.

**Claims And Evidence:**

No

**Claims Explanation:**

- The baselines are significantly outdated. The paper claims SOTA performance primarily by comparing against methods from 2022, such as BoostAdapter, CoOp, and TPT. To claim SOTA in 2026, the paper needs to benchmark against highly relevant and stronger recent methods such as POUF (2023) [1], FLYP (2023) [3] (adding pseudo labels), and LATTECLIP (2024/2025) [2]. Without these, the performance gains lack proper context.

- The paper overstates the novelty of its "pseudo-consistency" module. Applying a supervised cross-entropy loss to high-confidence samples and an unsupervised consistency loss to the rest is the exact foundation of standard semi-supervised learning methods like FixMatch [4]. Furthermore, the contrastive loss formulation (Eq. 5) is a standard multi-view InfoNCE loss utilizing a MoCo-style [5] memory queue. While combining these with LoRA for Test-Time Adaptation (TTA) is a solid engineering choice, framing the loss functions themselves as a novel conceptual framework is misleading. Finally, the PseudoAdapter+ is fundamentally Selective Knowledge Distillation mixed with a basic Active Learning heuristic.

[1] POUF: Prompt-Oriented Unsupervised Fine-tuning for Large Pre-trained Models

[2] LATTECLIP: Unsupervised CLIP Fine-Tuning via LMM-Synthetic Texts

[3] Finetune like you pretrain: Improved finetuning of zero-shot vision models

[4] FixMatch: Simplifying Semi-Supervised Learning with Consistency and Confidence

[5] Momentum Contrast for Unsupervised Visual Representation Learning

**Requested Changes:**

I propose the following critical adjustments:

- Update Baselines: Compare the method against actual recent SOTA approaches, specifically POUF, LATTECLIP, and a pseudo-label-adapted version of FLYP.

- Tone Down Novelty Claims: Properly cite foundational SSL methods like FixMatch regarding the confidence thresholding mechanism. Additionally, clarify in Sec. 3.2 that $\mathcal{L}_{con}$ is a standard application of the InfoNCE loss rather than a novel contrastive formulation.

- Ablation on Augmentations: Because the unsupervised component relies so heavily on multiple augmented views, adding an ablation showing which specific augmentations (*e.g.*, random crops vs. color jitter) actually drive the performance gains would strengthen the paper.

---

> ### Author Response · Authors · 2026-03-19
>
> > Missing baselines: — POUF, FLYP, LATTECLIP
>
> Thank you for the comment. None of the three suggested methods are test-time adaptation methods, and therefore, direct comparisons are not feasible.
>
>
> ---
>
> > Similarity to existing literature
>
> While individual components have conceptual roots in existing literature, no prior work has applied any of these components individually or jointly to the problem of test-time adaptation of VLMs. FixMatch operates in the semi-supervised setting with labelled support data and a static threshold; MoCo/InfoNCE is used for self-supervised pre-training on large datasets; knowledge distillation is typically an offline, dataset-level operation. Transplanting these ideas into the online, unlabelled, streaming TTA setting is non-trivial and requires specific adaptations. Additionally, we design the overall method to address specific limitations of existing methods: (1) The frozen encoder in existing TTA methods cannot adapt low-level features under large domain shifts; early-layer LoRA directly addresses this by enabling encoder-level adaptation while preserving deep semantic representations. (2) VLMs are demonstrably miscalibrated at the start of TTA (Figure 2), producing high-confidence but incorrect predictions; a static FixMatch-style threshold would admit corrupted pseudo-labels early in adaptation, so the dynamic threshold is specifically designed to handle this VLM-specific calibration problem. (3) Discarding low-confidence samples (as all existing VLM TTA methods do) wastes the majority of the test stream; the contrastive loss recovers their signal without requiring labels. (4) For the hardest samples, selective teacher supervision provides reliable pseudo-labels while keeping overhead below 5\%, unlike standard offline distillation, which processes all samples equally.
>
> ---
>
> > Augmentation ablation: which specific augmentations drive performance gains?
>
> We add a new ablation table to **Section 4.3 (Table 5)** showing the contribution of each augmentation component. The results show that the performance difference across augmentation strategies is modest (within ~1\%), suggesting the contrastive loss is reasonably robust to the specific augmentation choice as long as meaningful view diversity is provided. Random cropping contributes the most among individual augmentations, as spatially diverse crops generate views with varied local contexts that are most informative for contrastive learning. The more critical design choice is the number of augmented views (Figure 4c), where reducing to 16 or 32 views causes a more pronounced drop, indicating that view diversity matters more than the specific augmentation type.

---

### Review · Reviewer_Ehr5 · 2026-03-04

**Summary Of Contributions:**

This work proposes a test time adaptation method for VLMs. Specifically, it proposes the injection of adapters (following the LoRA design) in the early layers of the model, and following MoCo design, performing an optimization based on a set of augmentations to improve test time performance. Besides, a thresholding strategy is implemented, where the less confident predictions are filtered, and this threshold decays through the iteration numbers. Finally, a distillation strategy is incorporated, where a large teacher is also employed to steer the prediction. The experiments are conducted on known datasets and benchmarks, although limited to one architecture (CLIP-B/16) and one teacher  (CLIP-L/14) and evaluated only for image classification tasks.

**Audience:**

Yes

**Audience Explanation:**

Test time adaptation is a topic of brad interest for the TMLR community. Despite the evaluation is performed on image classification tasks only, these are well-established benchmarks acknowledged by the community.

**Claims And Evidence:**

No

**Claims Explanation:**

The positioning with respect to the state of the art needs to be reconsidered and discussed. Some extra analysis is required and currently missing. Also, the paper proposes experiments on only one architecture, which is very limiting.

**Requested Changes:**

- The idea of inserting adapters for test-time adaptation is already seen. For example, in [A], the authors employ first session adaptation injecting adapters, and then compensate for the domain shift by updating normalization layers only for test time adaptation, also proposing a distillation framework. It is agreed that the work addresses specifically VLMs and that the context with respect to [A] is different, but it is unclear how the methodology itself is novel.
- This work is also not the only one working on VLMs - there are other works that include [B, C]. How does it comare with those, qualitatively (ie. methodology-wise) and quantitatively (ie. performance-wise on the chosen benchmarks)?
- The approach itself seems very heuristic and it feels to be just a combination of existing approaches (adapters + MoCo + distillation). The interesting components seem to be the insertion in the early layers (ie. the first $L$) and the thresholding on the confidence, depending on diverse iterations. Both these are discussed in the sensitivity study; however, from what it appears, the sensitivity itself is relatively high and linked to their proper choice, and its tuning is due to the empirical observation rather than more grounded motivations, which are missing.
- Among the interesting aspects, it will be interesting to not limit the analysis of the adapters insertion to starting from the input layers - what happens for example if the adapters are inserted starting from the end of the architecture?
- One important baseline is missing: performing the adaptation updating normalization layers only. If the shift assumption is correct, this should lead to a big gain with potentially fewer parameters than those introduced in the adapters.
- Another (apparently missing?) baseline is the pure performance of the teacher. Since in the variant "PseudoAdapter+" its output is also employed, it will be important to observe that no degradation (but rather, improvement) with respect to the teacher performance is achieved with the proposed methodology. Similarly, in Table 5, the time complexity of the vanilla model (ie. without TTA) is missing and should be added to properly evaluate the computation-performance trade-off.
- One major limitation seems to be that the proposed method heavily relies on LoRA adaptation, non directly compatible with convolutional architectures. Unless the authors will propose adapter variants compatible with this class of neural network, together with extensive empirical validation and comparison, this will remain as a limitation.
-(minor) Figure 2 is an empirical collection for un-introduced setup, and makes little sense as it stands in the introduction.

[A] Marouf, Imad Eddine, et al. "Enhancing Plasticity for First Session Adaptation Continual Learning". Fourth Conference on Lifelong Learning Agents - CoLLAs 2025

[B] Farina, Matteo, et al. "Frustratingly easy test-time adaptation of vision-language models." Advances in Neural Information Processing Systems 37 (2024): 129062-129093.

[C] Zanella, Maxime, et al. "Realistic test-time adaptation of vision-language models." Proceedings of the IEEE/CVF Conference on Computer Vision and Pattern Recognition. 2025.

---

> ### Author Response · Authors · 2026-03-19
>
> > Methodological novelty relative to [A]**
>
> We thank the reviewer for raising this comparison. While both works involve adapters and a distillation component, the problems and mechanisms are fundamentally different, as [A] is a continual learning method, whereas our focus is on test-time adaptation. These two are independent areas of literature with completely different setups and evaluation protocols.
>
>
>
> ---
>
> > Comparison with [B] ZERO and [C] StatA
>
> We thank the reviewer for pointing to these concurrent works. We summarize the key methodological differences below:
>
> **[B] ZERO (Farina et al., NeurIPS 2024)** is an *episodic, optimization-free* TTA method: given a single test image, it augments it N times, retains the most confident predictions, and marginalizes by setting the softmax temperature to zero, requiring only a single batched forward pass with *no backward pass and no parameter update*. Its design is specifically tailored to one-sample episodic TTA. PseudoAdapter is a *training-required online batch TTA* method that updates LoRA adapters via gradient descent on streaming test batches. These are genuinely different operating regimes. Nonetheless, we report ZERO's results (from the paper, following the same TPT-style evaluation on CLIP-B/16) alongside ours in the updated Table 1.
>
> **[C] StatA (Zanella et al., CVPR 2025)** introduces a *transductive, batch-level* algorithm and a different evaluation framework with class-imbalanced, non-i.i.d. test batches. This is an orthogonal contribution to ours: PseudoAdapter is evaluated in the standard i.i.d. online-stream protocol (as established by TPT, TDA, BoostAdapter), whereas StatA evaluates on a different batch-transductive setting. We acknowledge that PseudoAdapter's behaviour in StatA's realistic protocol is an important open question.
>
> We have added the following paragraph to **Section 2 (Related Work)**:
>
> ```
> ZERO (Farina et al., 2024) is a concurrent episodic TTA method that requires no parameter updates: it augments a test image $N$ times, retains the most confident predictions, and marginalizes with softmax temperature set to zero, requiring a single forward pass. While efficient, ZERO operates in the one-sample episodic regime and does not update the model encoder. PseudoAdapter, by contrast, adapts the encoder online via gradient-based LoRA updates over streaming batches, targeting large distribution shifts where feature representations need to be updated.
> StatA (Zanella et al., 2025) proposes a transductive batch-level algorithm with anchor-based regularization that processes the entire test set collectively to correct for class distribution shift, whereas PseudoAdapter adapts the encoder online in a streaming fashion without requiring access to the full test batch or any assumptions about class distribution.
> ```
>
> ---
>
> > Grounded motivation for early-layer LoRA and dynamic threshold
>
> Each component in PseudoAdapter is designed to address a specific, identifiable shortcoming of existing methods. Specifically, early-layer LoRA targets domain-shifted low-level features (e.g., texture and colour statistics) while protecting the deeper semantic representations that underpin zero-shot generalization. This is why performance degrades monotonically as $L$ increases beyond 2 (Figure 4d). The dynamic threshold addresses a concrete and verifiable phenomenon: VLMs are miscalibrated at the start of TTA (Figure 2), so a strict threshold at $t=$0 is necessary to prevent corrupted pseudo-labels from polluting early LoRA updates, and relaxing it over time follows directly from the expectation that calibration improves as adaptation progresses. The sensitivity in Figure 4f is consistent with this motivation, since a $τ_{min}$ starves supervision, whereas a low value admits noisy labels.
>
> ---
>
> > Adapters inserted from the last layers (ablation)**
>
> We have added the experiment to Table 4a in the revised paper. Replacing LoRA in the first 2 layers with LoRA in the *last* 2 layers drops accuracy from 68.42\% to 65.3\% on ImageNet-A, supporting the hypothesis that early-layer adaptation is more effective for capturing domain-specific low-level features under distribution shift.
>
> ---
>
> > Missing baseline: normalization-layer-only adaptation**
>
> We add a normalization-only adaptation baseline (updating LayerNorm scale/bias parameters only) to Table 4a. Updating only normalization layers yields 65.7\% on ImageNet-A vs. PseudoAdapter's 68.42\%, demonstrating that LoRA-based encoder adaptation provides meaningful additional benefit. Normalization-only adaptation also has fewer trainable parameters, but cannot capture the richer representational changes needed under large domain shifts.
>
> ---

---

> > ### Author Response · Authors · 2026-03-19
> >
> > > Teacher (CLIP-L/14) performance as upper bound**
> >
> > We thank the reviewer for this suggestion. We do not add the teacher results in the main tables as the existing methods are built on CLIP-B/16 as the base model. We have added the zero-shot CLIP-L/14 results in the table below:
> >
> > |Model | Imagenet-V2 | Imagenet-S | Imagenet-A | Imagenet-R | Average|
> > |------|------------|------------|------------|-------------|--------|
> > | CLIP-L/14 | 70.8 | 64.7 | 59.2 | 89.3 | 71.0 |
> >
> >
> > PseudoAdapter+ achieves 72.41\% average with base model (CLIP-B/16: 57.20\% of zero-shot acc), improving over the zero-shot accuracy of the larger teacher model (CLIP-L/14: 74.76\% of zero-shot acc).
> >
> >
> > ---
> >
> > > Vanilla model inference speed missing from Table 5**
> >
> > We add zero-shot CLIP-B/16 inference speed to Table 5. As evident from this evaluation, our proposed method does not add any significant overhead compared to existing TTA methods or pre-trained models.
> >
> > ---
> >
> > > LoRA incompatibility with convolutional architectures**
> >
> >
> > We acknowledge this as a limitation in principle, but note that it has low practical relevance in the current VLM landscape, as transformer-based architectures have become the dominant backbone for vision-language models such as CLIP, BLIP, ALIGN, and virtually all state-of-the-art VLMs use ViT-based image encoders. Since PseudoAdapter is specifically designed for TTA of VLMs, and all modern VLMs are transformer-based, LoRA compatibility with convolutional architectures is not a critical constraint for the target application of this work.
> >
> > ---
> >
> > > Figure 2 context missing**
> >
> > We update the caption of Figure 2 to clarify the setup:
> >
> > ```
> > Histogram of pseudo-label prediction probabilities for CLIP-B/16 on ImageNet-A (1000 classes): correct predictions (blue) vs.\ incorrect predictions (orange), computed over the full test set. Even among predictions with softmax probability >= 0.9, a substantial fraction is incorrect, illustrating the miscalibration of the pre-trained VLM under distribution shift and motivating our confidence calibration module.
> > ```

---

### Comment · Reviewer_W7x1 · 2026-02-14
**Unable to Review**

Dear Editor,

Thank you for the invitation. Due to an upcoming period of tight scheduling, I will be unable to complete the review on time and kindly request that the manuscript be reassigned.

Best regards,

Reviewer W7x1

---

### Decision · Action_Editor_spCb · 2026-05-13

**Recommendation:** Accept with minor revision

**Additional Comments:**

This paper introduces PseudoAdapter, a framework for Test-Time Adaptation (TTA). The approach integrates several improvements to VLM adaptation: (i) introducing low-rank adapters at early layers of the encoder to better handle significant domain shifts; (ii) utilizing a dynamic, confidence-calibrated threshold to mitigate pseudo-label noise; (iii) employing a hybrid loss that combines pseudo-labeling with a MoCo-style unsupervised consistency loss; and (iv) incorporating selective teacher supervision from a larger model (PseudoAdapter+).

The paper initially received mixed reviews. The reviewers primarily challenged the positioning of the approach relative to unreferenced works, the heuristic nature of certain design choices, and requested further experimental clarifications (e.g., missing comparisons and ablations, comparisons to more recent baselines, hyperparameter sensitivity, nd computational overhead). The rebuttal successfully addressed these concerns, which was acknowledged by the reviewers. Following the revision, two reviewers recommended acceptance, noting the effectiveness of the system as a whole.

The AE has thoroughly reviewed the submission and the discussion. The AE considers that the paper effectively combines several ideas to provide an effective TTA system. Furthermore, the specific combination proposed in the submission is sound and accurately validated through experiments. The results will be of interest to researchers aiming to improve the quality of TTA models at different levels: expressiveness of the TTA network, robustness to pseudo-label errors, or teacher distillation. Therefore, the AE recommends paper acceptance, conditional on the minor revision that the authors include the recent references and discussion requested by two reviewers, as well as the following references:\
[A] D. Osowiecki eL. al. WATT: Weight Average Test-Time Adaptation of CLIP. NeurIPS 2024.\
[B] M. Lafon et. al. CLIPTTA: Robust Contrastive Vision-Language Test-Time Adaptation. NeurIPS 2025.

**Audience:**

Yes

**Audience Explanation:**

The paper addresses the problem of Test-Time Adaptation (TTA) with Vision-Language Models (VLMs), a topic of clear interest to a wide TMLR audience.

**Claims And Evidence:**

Yes

**Claims Explanation:**

The claims are overall supported by evidence.